# Bayesian Comparisons Between Representations

**Heiko H. Schütt** (heiko.schutt@uni.lu)

DBCS, Université du Luxembourg, 2, place de l'Université, L-4365 Esch-sur-Alzette, Luxembourg

## Abstract

**Which neural networks are similar is a fundamental question for both machine learning and neuroscience. Here, it is proposed to base comparisons on the predictive distributions of linear readouts from intermediate representations. In Bayesian statistics, the prior predictive distribution is a full description of the inductive bias and generalization of a model, making it a great basis for comparisons. This distribution directly gives the evidence a dataset would provide in favor of the model. If we want to compare multiple models to each other, we can use a metric for probability distributions like the Jensen-Shannon distance or the total variation distance. As these are metrics, this induces pseudo-metrics for representations, which measure how well two representations could be distinguished based on a linear read out. For a linear readout with a Gaussian prior on the read-out weights and Gaussian noise, we can analytically compute the (prior and posterior) predictive distributions without approximations. These distributions depend only on the linear kernel matrix of the representations in the model. Thus, the Bayesian metrics connect to both linear read-out based comparisons and kernel based metrics like centered kernel alignment and representational similarity analysis. The new methods are demonstrated with deep neural networks trained on ImageNet-1k comparing them to each other and a small subset of the Natural Scenes Dataset. The Bayesian comparisons are correlated to but distinct from existing metrics. Evaluations vary slightly less across random image samples and yield informative results with full uncertainty information. Thus the proposed Bayesian metrics nicely extend our toolkit for comparing representations.**

**Keywords:** Representations, Comparisons, Bayesian, RSA, CKA, linear encoding models

## Introduction

For both machine learning and neuroscience, a fundamental question is which neural networks are similar to each other. The deep learning revolution has brought about a broad range of networks which enable an equally broad range of machine learning applications and are used as models of biological neural networks. To evaluate these models and to compare them to each other requires good formal methods. On the machine learning side (Kornblith et al., 2019; Williams et al., 2021), measures of similarity are used to judge which architectural changes, training parameters, and similar aspects have an influence on the processing (e.g. Neyshabur et al., 2020; A. Raghu et al., 2019; M. Raghu et al., 2021), to train models to perform similar to an existing model in a teacher-student setup (Wang & Yoon, 2022; Passalis & Tefas, 2018), and for visualization of the processing through a network (Williams et al., 2021). On the neuroscience side, it is a central question whether the processing in any given model is similar to human processing or not. It has been studied extensively how to measure similarity best (Diedrichsen et al., 2011; Diedrichsen & Kriegeskorte, 2017; Khaligh-Razavi et al., 2017; Naselaris et al., 2011; Storrs et al., 2021; Schütt et al., 2023). However, this discussion is not settled yet and popular competitions use different metrics to compare models to brains (e.g. Cichy et al., 2019; Hebart et al., 2019; Schrimpf et al., 2020).

Here, we explore a new Bayesian approach to the comparison of representations. Bayesian statistics are particularly well suited for this situation, because they deal better than frequentist statistics with small datasets and parameters that are not strongly constrained by the data. For high dimensional representations in deep neural networks, all datasets we can possibly use are small in this statistical sense such that this advantage applies to almost all possible applications.

For a ridge regularized linear readout model, we can directly compute the prior predictive distribution, which turns out to be a normal distribution with a computable covariance between predictions for different inputs. This predictive distribution is a full description of the inductive bias and generalization behavior of this linear readout model. When we have neural data available, the predictive distributions allow us to directly evaluate model evidence for full Bayesian inference for model selection. By using a metric on the predictive predictions, we get pseudo-metrics for representations, which characterize how well representations could be distinguished based on linear readouts from them. First, the new analysis methods are described and analyzed theoretically and then applied to commonly used image processing models trained on ImageNet-1k for further evaluation.

## Related Work

Here, we deal with the general question how to compare neural networks to each other. For this purpose, a neural network is a series of parameterized functions that are applied to an input and the outputs of previous functions. We call the outputs of these functions representations of the input. Different networks use different functions to create representations and generally have no corresponding parts or parameter values. And for biological neurons, we typically only record their activities, which do not correspond to weights or function parameters. Thus, most comparisons are based on comparing representations rather than specifications of the functions themselves (Diedrichsen & Kriegeskorte, 2017). There are three major approaches to comparing models and their representations.

The first type of comparisons is based on comparing only

the output of models. The immediate appeal of this approach is that all models for a specific task need to make predictions in the same format. Thus, models can be compared at this stage independent of their internal computations. The first metric for comparison is overall performance. For detailed comparisons, an analysis of the errors can be more informative (Geirhos, Janssen, et al., 2018; Geirhos, Temme, et al., 2018). To enable comparisons of internal representations one can train a (usually linear or logistic) "probe" that predicts something based on the internal representation (Alain & Bengio, 2016). The results depend on the details of the evaluation task and on the training scheme for the readout. To reduce the dependence on the exact task, a recent method searches for the task with the most different results (Boix-Adsera et al., 2022).

The second type of comparisons is based on (linearly) mapping one of the models to the other, which is known in neuroscience as an encoding model (Naselaris et al., 2011). Comparisons based on fitting an explicit map are asymmetric by default. For some situations like predicting brain measurements based on a model representation, this is sensible. For comparisons between model representations, machine learning favors symmetric measures like variations of canonical correlation analysis (Hardoon et al., 2004; M. Raghu et al., 2017). Ideally, we should aim for a metric though, such that our intuitions about similarity hold and clustering or embedding methods work effectively. This can be achieved by using generalized shape metrics (Williams et al., 2021).

The third type of comparisons is based on similarity structure. The first step of this approach is to compute a matrix of pairwise (dis-)similarities between stimuli or conditions for each model. These matrices can be compared directly. In neuroscience, this is known as representational similarity analysis (Kriegeskorte et al., 2008). In machine learning, this is based on the Kernel (similarity) matrix instead of the dissimilarity matrix and such methods are known as centered Kernel alignment (Kornblith et al., 2019). The neuroscience and machine learning approaches are similar and, in some cases, exactly equivalent (Diedrichsen et al., 2021).

There are deep connections between these different approaches (Diedrichsen & Kriegeskorte, 2017; Harvey et al., 2023) and a recent paper links them to decoding tasks Harvey et al. (2024). Indeed, the Bayesian metrics proposed here connect to all three branches: They are (2) based on linear probes for (1) the tasks and the distributions we handle depend only on the (3) kernel matrix of the representations of the inputs.

In machine learning, distances for probability distributions are regularly used for probabilistic models Neal (1996); Murphy (2022), in particular for training and comparing generative models (e.g. Li et al., 2016; Gretton et al., 2012)

## Methods

### Bayesian comparison framework

We will apply Bayesian statistics to the (typically linear) models that are used to map representations to a task output (Fig. 1). Once we define a prior distribution over the weights, each representation then predicts a joint distribution of outputs for any set of stimuli. Formally, for a list of stimuli $S = [s_1, \ldots, s_n]$ represented by vectors $x_i$, and a potentially random readout function $f_\theta$ from the space of representations into the space of outputs the probability of observing a specific vector of outputs $y$ is:

$$p(y|S) = \int p(f_\theta(x) = y)p(\theta)d\theta \qquad (1)$$

This distribution directly gives the probability of any dataset under the model, which is what we need to evaluate a model according to Bayesian statistics. To compare models to each other, we can apply a distance for probability distributions to the predictive distributions of the models. Such distances quantify how well the two distributions can be separated based on a random draw from them, i.e. how well one could tell which of the two models a random linear readout comes from. Additionally, the prior predictive distribution is a complete description of the generalization behavior of the model, i.e. of the inductive bias for any type of task a read-out model might be trained for on the test stimuli.

### Linear readout models

While the Bayesian Framework in theory supports any set of read out functions with a prior over them, we focus on linear read outs with a isotropic Gaussian weight prior here. This setup is the Bayesian treatment for ridge regression with a fixed regularization strength.

Formally, the predictive distribution for a mean readout $\bar{\mathbf{y}} \in \mathbb{R}^n$ for $n$ stimuli with representations $\mathbf{x}_i \in \mathbb{R}^k$ concatenated in a matrix of representations $X \in \mathbb{R}^{k,n}$ with rows for each stimulus and columns for each feature dimension in the model is:

$$\bar{\mathbf{y}} = X\beta \qquad \beta \sim N(0, \sigma_\beta^2 I) \qquad (2)$$

As linear transformations of Gaussians are Gaussian, the distribution for $\bar{\mathbf{y}}$ is then also a Gaussian with known parameters:

$$\bar{\mathbf{y}} \sim N(0, \sigma_\beta^2 XX^T) \qquad (3)$$

With many stimuli or low-dimensional representations, $XX^T$ can become rank deficient such that the predictive distributions for the mean are degenerate. The natural solution for this problem is to take into account that measurements for a readout will be noisy and add independent Gaussian noise with standard deviation $\sigma_\epsilon$ to each observation. The predictive distribution for observations $\mathbf{y}$ then is:

$$\mathbf{y} \sim N(0, \sigma_\beta^2 XX^T + \sigma_\epsilon^2 I) \qquad (4)$$

To properly fit a linear readout one needs to adjust $\sigma_\beta$ to induce the right amount of regularization. When implementing this, we can compensate for any re-scaling of $X$ or $XX^T$ by adjusting $\sigma_\beta$. For comparing models, we should thus ignore the overall scale of $X$. To do so, $\sigma_\beta$ is set such that the trace of the covariance matrix $\mathrm{tr}\left(\sigma_\beta^2 XX^T\right)$ is $n$, i.e. $\sigma_\beta^2 = n\,\mathrm{tr}^{-1}\left(XX^T\right)$. Setting the trace to any other value by multiplying the $\sigma_\beta^2$ for all

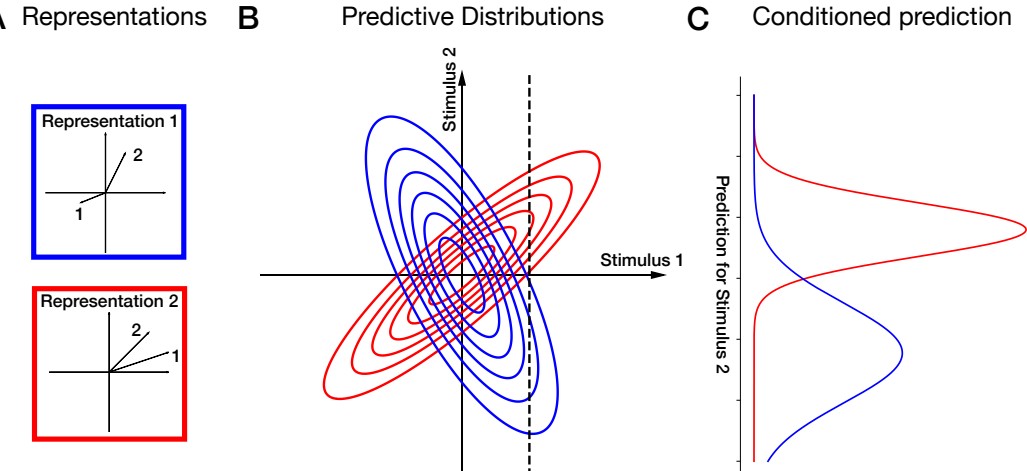

**A** Representations  **B** Predictive Distributions  **C** Conditioned prediction

Figure 1: Minimal example for the Bayesian comparison framework: two stimuli in two 2D representations **A**: The original representations of the two stimuli. **B**: Predictive Distributions induced by a linear read out model with a zero mean Gaussian weight prior. These are the distributions we compare to determine the (dis-)similarity of representations. **C**: Prediction for Stimulus 2 according to the two models if a value of 1 for Stimulus 1 is given as training data. In the Bayesian statistics these are computed by conditioning the distribution in B, corresponding to the cut at the dashed line through a value of 1 for Stimulus 1.

models by the same constant would yield the same distances according to the probability distribution metrics used here.

The noise variance $\sigma_\epsilon^2$ is a property of the measurement which can be fit and kept constant for evaluations on measured data. For comparisons between models, we will keep all our distributions normalized to an average variance of 1 per stimulus. This leaves us with one free parameter $a \in [0,1]$ to trade off signal and noise variance, which yields the following distribution for the observations:

$$\mathbf{y} \sim N\left(0, (1-a)\frac{nXX^T}{\text{tr}(XX^T)} + aI\right) \quad (5)$$

A heuristic for $a$ depending on the number of images to yield variation in the distances is given below. To justify this, I need to define the distances between distributions to use first though.

**Distances between predictive distributions**

Here, two metrics for probability distributions are used to measure how similar the predictive distributions based on representations are: the Total Variation Distance (TVD) and the Jensen-Shannon Divergence (JSD) (Endres & Schindelin, 2003). As there are no closed form solutions for computing these distances between Gaussians, they are approximated based on $N = 10\,000$ draws from the respective distributions. For this section, the two predictive distributions to be compared are called $P_1$ and $P_2$ with densities $p_1$ and $p_2$.

**Total variation distance** (TVD) is proportional to the accuracy of the optimal hard classifier to distinguish samples from

the two distributions and is defined as:

$$\text{TVD}(P_1, P_2) = \sup_A |P_1(A) - P_2(A)| \quad (6)$$

where $A$ can be measurable set.

For continuous distributions like the Gaussians we deal with here, there are always at least two equivalent $A$ that maximize the difference: $A_1 = \{x : p_1(x) > p_2(x)\}$ and $A_2 = \{x : p_2(x) > p_1(x)\}$. For each of these, we can generate an approximation for the TVD based on samples from $P_1$ or $P_2$ respectively: We first note that the density with respect to $P_1$ of $P_1$ is 1 and of $P_2$ is $\frac{p_2}{p_1}$. The difference in probabilities for the event that $p_1 > p_2$ is thus $\int \max\left(0, 1 - \frac{p_2(x)}{p_1(x)}\right) dP_1(x)$, which we can approximate with the standard sampling approximation with samples from $P_1$. A completely analogous derivation yields an approximation for the probabilities of $A_2$ based on samples from $P_2$. Averaging the two approximations yields the following approximation of the $TVD$ that was used for all computations of the TVD:

$$\text{TVD}(P_1, P_2) \approx \frac{1}{2N} \sum_{i=1}^{N} \max\left(0, 1 - \frac{p_2(x_i^{(1)})}{p_1(x_i^{(1)})}\right) \quad (7)$$

$$+ \frac{1}{2N} \sum_{i=1}^{N} \max\left(0, 1 - \frac{p_1(x_i^{(2)})}{p_2(x_i^{(2)})}\right) \quad (8)$$

where $x^{(1)}$ is a sample of size $N$ from $P_1$ and $x^{(2)}$ is a sample of size $N$ from $P_2$.

**Jensen Shannon Divergence** (JSD) gives the mutual information between the draws from the distribution and the label

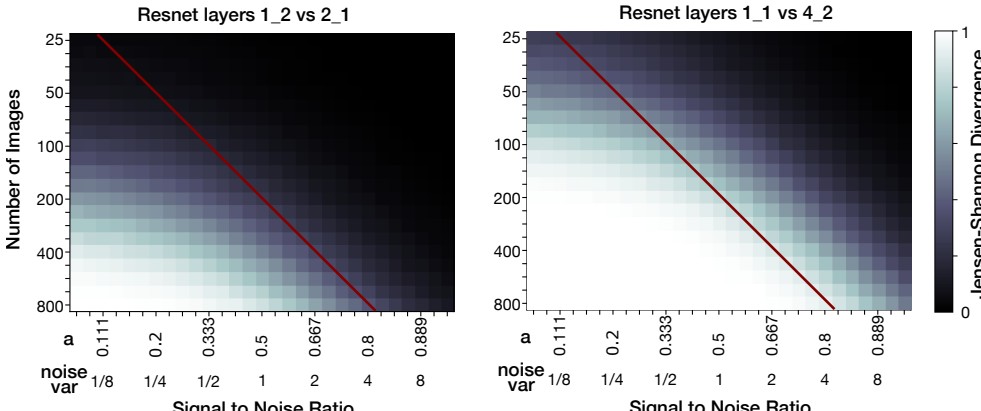

Figure 2: Dependence of the Jensen Shannon Divergence (JSD) on the number of images used and the signal to noise ratio for two comparisons within a standard ResNet-18. For the signal to noise ratio, two labels are shown: the noise variance for signal variance 1 and the mixture factor $a$ as defined in the text. The red line shows the slope such that the noise variance is proportional to the number of images. Note that JSD is fairly constant along this line once enough images are collected, while JSD gets small for few images independent of the Signal to noise ratio. Left: Divergence between two close representations—the outputs of the first layer and the output of the fist block of the second layer. Right: Divergence between two different representations—the first block in the first layer and the last block in the last convolution layer.

which distribution the sample came from. It is defined as:

$$\text{JSD}(P_1, P_2) = \int p_1(x) \log_2 \frac{p_1(x)}{p_1(x) + p_2(x)} dx \tag{9}$$

$$+ \int p_2(x) \log_2 \frac{p_2(x)}{p_1(x) + p_2(x)} dx - 1, \tag{10}$$

where the $\frac{1}{2}$ was pulled out of the denominators and $\log_2$ was used, such that the JSD ranges from 0 to 1. The Jensen Shannon Distance is the square root of this value, which is a metric (Endres & Schindelin, 2003). The two integrals can be approximated with samples from $P_1$ and $P_2$ respectively, which yields the following approximation used throughout this paper:

$$\text{JSD}(P_1, P_2) \approx \frac{1}{N} \sum_{i=1}^{N} \log_2 \frac{p_1(x_i^{(1)})}{p_1(x_i^{(1)}) + p_2(x_i^{(1)})} \tag{11}$$

$$+ \frac{1}{N} \sum_{i=1}^{N} \log_2 \frac{p_2(x_i^{(2)})}{p_1(x_i^{(2)}) + p_2(x_i^{(2)})} - 1 \tag{12}$$

where $x^{(1)}$ is a sample of size $N$ from $P_1$ and $x^{(2)}$ is a sample of size $N$ from $P_2$.

**Gradient** Both distance estimates are based on a sum over samples from the two Gaussian distributions. To get a gradient for this function, we can use the reparameterization trick, i.e. we take standard normal samples which are transformed to have the right covariance matrix such that our approximation becomes a differentiable function of the covariance matrices and these random samples, allowing us to compute a gradient through the distance computations(Kingma & Welling, 2013).

## Choosing the signal to noise ratio

When we use more images, representations become easier to discriminate and both TVD and JSD grow (Fig. 2). This makes comparisons with many images uninformative, because all representations become perfectly discriminable and have distances close to 1 to each other. To prevent this, we can adjust the mixture weight for the noise $a$ to compensate for the number of stimuli.

A sensible dependence between $a$ and the number of stimuli $n$ can be derived if we assume that the variance of the noise is proportional to the number of images used. This corresponds to the scaling we get if we repeat measurements of the few images such that we take the same overall number of measurements independent of the number of images and assume the usual $1/n$ relationship for the noise variance. This yields $a = \frac{bn}{1+bn}$ where $b$ is a constant that makes our analysis overall more or less sensitive.

This adjustment of $a$ does indeed yield a relatively constant level of discriminability once enough images are used for testing (Fig. 2). For small image numbers discriminability tends to fall off independent of the noise variance, even for completely noise free predictions. Based on a few examples, I settled on $b = 1/100$ such that 100 images yield $a = 0.5$ for my illustrations. More fine-grained distinctions may profit from using lower noise levels and broader distinctions from even higher noise levels.

## Pseudo-metric

For further analyses of the similarities between representations it is advantageous if the similarities are a pseudo-metric on the space of representations (Williams et al., 2021), because we can then guarantee the convergence and performance of embedding, clustering and other analysis methods.

The new measures of similarity between representations are pseudo-metrics, because we use metrics to compare the predictive distributions. As we have a map from the representation to its predictive distribution this induces a pseudo-metric on the representations, but not a metric, because multiple representations map to the same predictive distribution.

### Equivalent representations

Considering some representations to be equivalent is generally desirable, because some representations are indeed completely equivalent. For example, two representations that contain the same features in different orders should be considered the same. There is some discussion on what transformations one should ignore when comparing representations though (Williams et al., 2021; Kriegeskorte & Diedrichsen, 2019).

To understand what the new metrics measure, it is informative to consider which representations are equivalent according to it, i.e. have distance 0 between them. As TVD and JSD are metrics on probability distributions, representations are equivalent if and only if their predictive distributions are the same[1]. For the 0-mean normal distributions we are comparing here, this is equivalent to their covariance matrices being equal. Thus, two representations $\phi$ and $\psi$ are equivalent, iff:

$$\frac{X_\psi X_\psi^T}{\text{tr}(X_\psi X_\psi^T)} = \frac{X_\phi X_\phi^T}{\text{tr}(X_\phi X_\phi^T)} \quad (13)$$

The normalization to total variance 1 maps all representations that are scaled by a constant $k \in \mathbb{R} \backslash \{0\}$ to the same covariance. Thus, representations that differ only by multiplication with a constant are equivalent. Additionally, any unitary or rotation matrix $U$ such that $I = UU^T$ applied to the features will yield the same covariance matrix. Formally, if one representation $X' = XU$ is a rotation of a representation $X$, then $X'X'^T = XUU^TX^T = XX^T$, i.e. the representations induce the same covariance and are equivalent.

The new measures do not ignore the norms of the individual patterns or their distance to the origin. The norms determine the predicted variances for the individual stimuli and after normalization the relative sizes of variances are preserved. This is in contrast to representational similarity analysis (Kriegeskorte et al., 2008) which analyses only differences between representations and centered kernel alignment, which explicitly removes this information by centering (Kornblith et al., 2019). This also implies that the Bayesian measures are not invariant to shifts of the representation, i.e. to adding an offset to all representation vectors. This is in contrast to rotation invariant generalized shape metrics, which contain a centering operation which makes them invariant to shifts (Williams et al., 2021). Thus the Bayesian measures are stricter than these metrics.

### Experiments

I evaluated the Bayesian methods by comparing deep neural network representations from ImageNet-1k trained models provided with the *torchvision* (maintainers & contributors, 2016)

[1] Except for a subset of measure 0.

package in python. As test images, I used randomly chosen unlabeled images from MS COCO (Lin et al., 2015) form their 'unlabeled2017' folder with a single center crop and the preprocessing required by the respective models. All experiments reported here were run on a single MacBook Pro, M2max with 96Gb of RAM. Single distances are usually computed within less than a second and the longest experiment took 67 minutes of computation time in total. See Appendix for more details.

## Applications & Results

### Example

As an example application, I analyzed the similarities between intermediate representations from 3 standard neural networks based on the Jensen-Shannon-Distance between linear predictions proposed here (Fig. 3). AlexNet (Krizhevsky et al., 2012), ResNet-18 (He et al., 2016) and the Vision Transformer B-16 (ViT-B-16) (Dosovitskiy et al., 2021) were obtained from torchvision (maintainers & contributors, 2016) with the standard weights from ImageNet-1k training. The analysis is based on 200 images from the unlabeled set of MS COCO images and I chose $a = 2/3$ based on the heuristic above (see Appendix for more details). For convolutional layers the tensors were simply flattened, i.e. each location was considered a separate dimension of the representation with its own weight.

The Bayesian metrics yield sensible results. Nearby layers produce similar representations (Fig. 3A), such that the processing of each network forms a relatively smooth path through the space of representations (Fig. 3B). The distances fill the full range from 0 to 1 and intermediate representations in different networks show some similarities. I display only the Jensen-Shannon-Distance here, because the two proposed metrics turn out to be very similar in the next section.

Analyses like these can be very informative for understanding how networks process their inputs. The Vision Transformer (ViT-B-16) is a particular interesting example, because its layers all have the same internal architecture, while there is a substantial break in terms of functional similarity that splits the layers into two clusters of representations, which have no correspondence to any break in the architecture.

### Comparison to other metrics

To compare the two metrics to each other and to other measures of dissimilarity of neural network representations, all layers of AlexNet and ResNet-18 were compared to each other (Fig. 4). 100 random samples of 100 images each from the first 1000 unlabeled images of MS COCO were used as test stimuli. The analysis is based on the mean across image samples (more details in Appendix ). In broad strokes the metrics all agree: Close-by neural network layers tend to be similar to each other; some of the intermediate convolutional layers produce similar representations in the two networks; and the final readout layers of both networks are very different from the convolutional layers.

The TVD yields extremely similar results to the JSD (Pearson correlation $r(198) = 0.99980$). They are not exactly the same

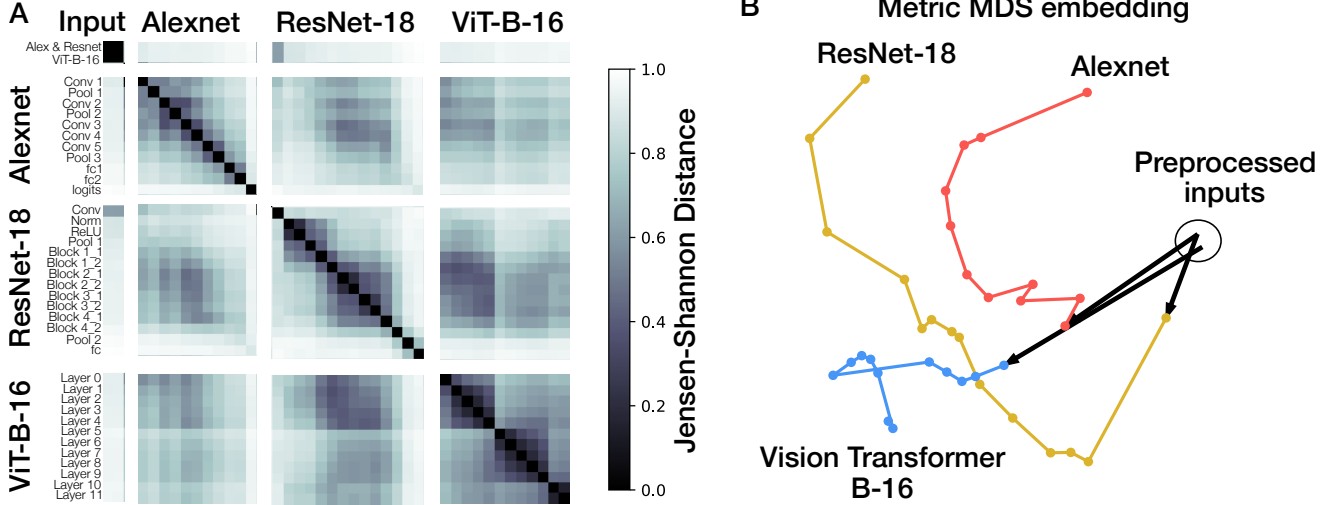

Figure 3: Example analysis based on the Jensen-Shannon Distance as proposed here. A range of layers from AlexNet, ResNet-18 and the Vision Transformer B-16 (ViT-B-16) are compared based on 200 randomly chosen natural images. Weights for all networks were obtained from torchvision and were originally trained on ImageNet-1k .**A**: Distance matrix according to the Jensen-Shannon Distance including the pre-processed input images. **B**: Metric MDS embedding of the layers into a 2D space with arbitrary units.

though. The Total Variation Distance is systematically ever so slightly smaller for intermediate values and this difference is larger than the numerical errors. Thus, for all intents and purposes the two metrics are equivalent, except for the Jensen-Shannon distance being close to the square root of the TVD. The other metrics also show fairly strong relationships with the newly proposed metrics. In particular, the arccos of the centered kernel alignment (Kornblith et al., 2019) as suggested by Williams et al. (2021) shows a fairly close relationship. The non-metric measures show bigger differences from the new metrics and RSA measures are least similar to the new metrics.

### Stability analysis

We need to characterize how variable the results of our analyses are to know how confident we should be in our results. The variability of our results depend on the amount of test data we use of course which complicates this judgment.

**Number of Images** Different test images will yield different distances between layers. This is probably the largest source of variability for noiseless representations. To understand how much variance this causes, I analyzed the repeats of the simulations used for comparisons to other metrics above (Tab. 1). The variance of estimates depends substantially on the mean distance value and the means are different for different image numbers and different metrics. For this reason, the maximum and median standard deviations appear to be more informative than mean standard deviations and we should nonetheless be careful when interpreting these values.

Nonetheless, we can make three observations: First, the standard deviations are reasonably small for most metrics and decrease at least as fast as expected ($1/\sqrt{n}$ with the number of

test images). Second, the new metrics are at least as stable as existing ones, perhaps a little more so. Third, the correlation distance of RSA varies far more than other metrics, by more than a factor 10 in variance. Thus, the Bayesian metrics are certainly competitive in terms of stability, but the correlation distance between representational dissimilarity matrices should not be used for comparisons between DNN layers.

**Number of samples** We need to approximate the distances based on sampling approximations to the integrals that occur. This creates an approximation error. We can estimate the size of this error from the variance of the averaged values (see App. for details). For both metrics, the dependence between variance and the true distance was independent of the number of stimuli and the dimensionality of the representations. It is inverse-u-shaped with the variance approaching 0 at 0 and 1 and the peak at about 0.6 for total variation distance and 0.5 Jensen-Shannon Divergence. The maximum variance is about $\frac{0.339}{N}$ for Jensen-Shannon Divergence ($SD(N = 10\,000) \approx 0.0058$) and about $\frac{0.071}{N}$ for total variation distance ($SD(N = 10\,000) \approx 0.0027$).

### Evaluation on neural data

To test the evaluation methods for neural data, I evaluated AlexNet and RestNet-18 on the natural scenes dataset (Allen et al., 2021) as prepared for the Algonauts challenge 2023 (Gifford et al., 2023).

First, the prior predictive distribution was used to evaluate all layers of the two networks at predicting voxel responses in the left hemisphere's PPA to the first 500 images (see Fig. 5A). This analysis yields a posterior over layers for each individual

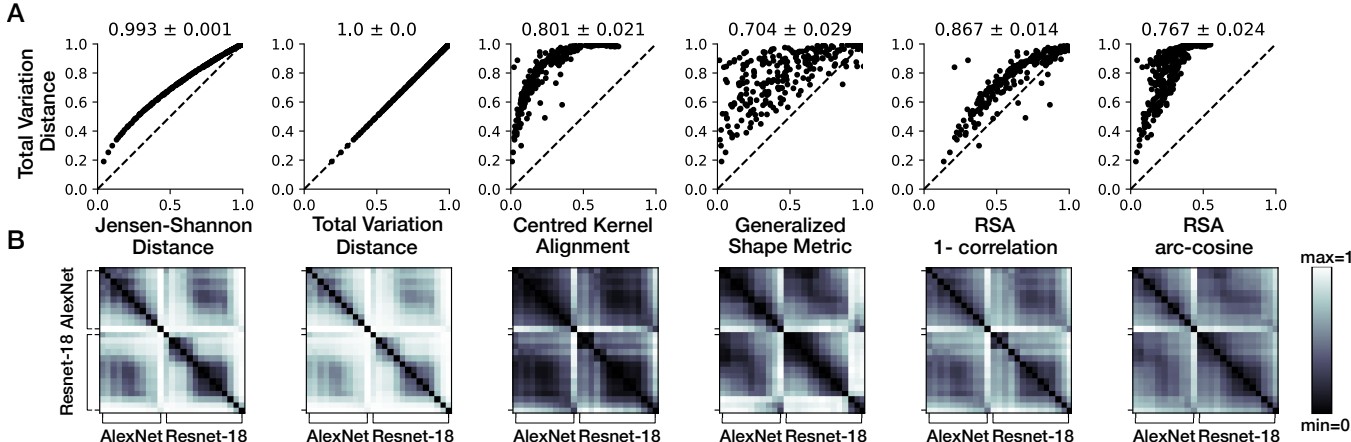

Figure 4: Comparisons between metrics based on all pairwise distances between layers of Alexnet and Resnet-18 using 100 random unlabeled images from MS COCO as inputs. **A**: Plotting different dissimilarity measures against the total variation distance proposed here. The number above each plot gives the Pearson correlation $\pm$ the coarse analytic estimate of the standard deviation $(1-r^2)/\sqrt{N-3}$ (Gnambs, 2023). **B**: The matrix of pairwise comparisons between layers according to the different dissimilarity measures. **Compared measures**: Jensen-Shannon-Distance & Total variation distance as proposed here. Centered Kernel Alignment: One minus the linear centered kernel alignment. Generalized Shape Metric: $\mathrm{arccos}$ of the centered kernel alignment, which is a shape metric (Williams et al., 2021, Appendix C.7). RSA 1-correlation: Representational similarity analysis based on one minus the Pearson correlation of euclidean distances. RSA arc-cosine: Representational similarity analysis based on the $\mathrm{arccos}$ of the cosine similarity of euclidean distances.

Table 1: Standard deviations of the distance measures across different image choices based on 100 random draws of n=25/50/100 images. Each cell shows maximum / median across the 300 distances among the 25 layers of AlexNet and ResNet-18 used for all comparisons. See Fig. 4 for details on the metrics.

| med / max | JSD | TVD | CKA | Shape Metric | RSA 1-corr | RSA arccos |
|---|---|---|---|---|---|---|
| $n = 25$ | 0.029/0.068 | 0.029/0.069 | 0.027/0.056 | 0.051/0.094 | 0.144/0.246 | 0.041/0.079 |
| $n = 50$ | 0.017/0.055 | 0.018/0.055 | 0.021/0.044 | 0.037/0.077 | 0.103/0.182 | 0.031/0.068 |
| $n = 100$ | 0.008/0.034 | 0.009/0.035 | 0.016/0.034 | 0.025/0.055 | 0.070/0.135 | 0.021/0.049 |

voxel. Based on this analysis only nine layers are the best model for any voxel in this subject's PPA. The best performing layers at predicting this area is the pooling layer from ResNet-18. This analysis demonstrates that we can run the proposed fully Bayesian analyses to choose among layers as models of visual cortex and get sensible useful results including a characterization of our uncertainty.

Second, we can look at how training data for the readout changes the predictions based on our network layers (see derivations in Appendix). Here, the predictions for the second 500 images in the dataset are compared based on the original weight prior to the predictions conditioned on the response to the first 500 images (see Fig. 5 B&C). As an example, I use the first voxel of PPA. The predictions are indeed different. In particular, the predictions of the models become more similar on average, due to the layers performing badly moving closer to the predictions that match the data well. Some of the better performing models become more dissimilar though, indicating that the prior knowledge of how the voxel responded to other images is helpful to distinguish models. This is expected behavior for the dissimilarities between layers.

We thus get all information we may want to get from comparing networks to brains: Which models match the data well, what our expected responses are for future images, how similar those predictions are for different models, and uncertainty information about all those results.

## Discussion

Here, new measures for the similarity of representations are presented, which are are based on a Bayesian analysis of linear readouts. When we compare to measured data, we can directly perform Bayesian inference. To compare models, JSD and TVD provide (pseudo-) metrics for representations, which quantify how well models can be discriminated. All measures can be computed from the linear kernel matrix of the representations with a stochastic gradient. We can use training data for a task to focus comparisons by comparing posterior predictive distributions instead of prior predictive ones.

Bayesian inference for the linear model is analytically tractable for up to $1\,000$s of stimuli, independent of the dimensionality of the representations until the Cholesky decomposition of covariance matrices over stimuli becomes compu-

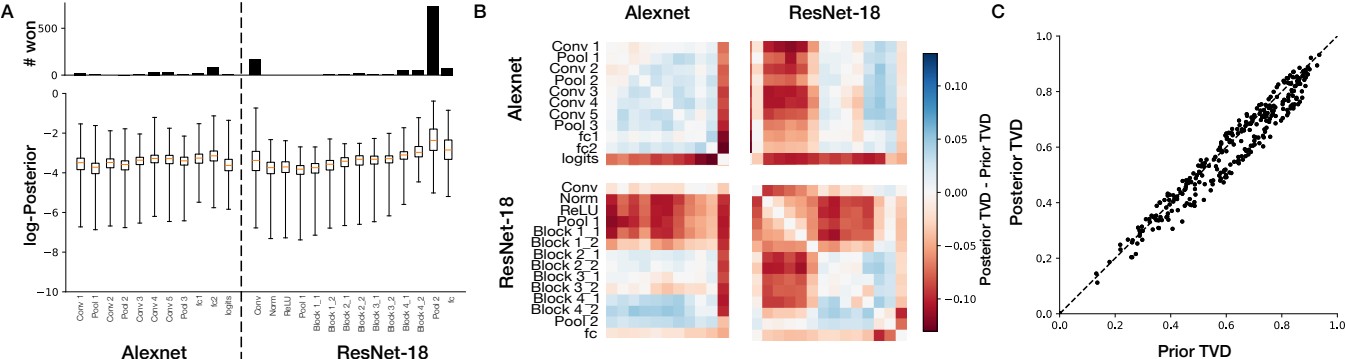

Figure 5: Evaluation of AlexNet and ResNet-18 layers at predicting fMRI voxels measured in left PPA for the natural scenes dataset (Allen et al., 2021) as prepared for the Algonauts challenge 2023 (Gifford et al., 2023) using the first 500 images as training and the next 500 for evaluation. **A**: Evaluation of the prior prediction on the training data. Top shows the number of voxels that each of the layers performed best for. Bottom shows the distribution of log-Posterior values for each layer, which is proportional to the evidence in favor of each layer. Clearly some of the layers could be excluded based on this data. **B**: Comparison of the posterior TVD (based on the distribution conditioned on the training data) to the prior TVD (based on the prior weight distribution) for the first voxel of PPA in the dataset. The biggest changes are for layers that perform badly, which become more similar to the ones that perform well, but some layers that perform similarly become more different. **C**: Same data as in B, but plotting the two TVDs against each other.

tationally intractable. This number is sufficient for most probe tasks and batch-sizes used for deep neural networks, but we cannot compute similarities based on complete deep learning datasets. Total variation distance and Jensen-Shannon distance give very similar results, but the numerical approximation is more stable and efficient for the total variation distance. Thus, the total variation distance should be the default choice.

A clear advantage of basing comparison methods on Bayesian statistics is that we can integrate our comparisons into a coherent fully probabilistic inference. This is in contrast to other comparison measures that require complex cross-validation and bootstrap procedures to allow valid statistical inference (Schütt et al., 2023). Removing these complications can make analysis faster and simpler, more powerful due to using the whole dataset for evaluation, and leaves fewer analysis choices to the researcher improving standardization. For simplicities sake, the Bayesian analysis presented here (Fig. 5 A) is based on a separate analysis per voxel for a single subject. Full inference about the whole dataset will require further development of adequate methods to pool over voxels and subjects, but seems within reach. Already today, the Bayesian analysis is faster and simpler than the frequentist encoding model analysis.

The connection to discriminability and the (stochastic) gradient make the Bayesian metrics ideal targets for stimulus optimization techniques (e.g. Golan et al. (2020, 2022)). The analysis provides exactly the information needed to implement such adaptive designs: Probabilistic inference about which models fit the existing data best and a measure how well a stimulus set separates a pair of models. Additionally analyses like the ones in Fig. 2 are effectively power analyses based on the chosen stimuli and the signal to noise ratio, which will be

helpful to check experimental designs more generally.

An avenue for future development are slightly more complicated models for individual measurement channels. One could consider a non-zero mean for the weights, which would imply that models predict a non-zero mean for the output and which stimuli yield higher outputs. Alternatively, an extension towards generalized linear models could be interesting to capture recordings from spiking neurons or other non-Gaussian data and tasks that require such predictions.

All results in this paper concern image classification models as this is my area of expertise and among the most common types of deep neural network. The methods are not inherently restricted to image processing models in any way though. As long as we can process a set of different inputs and think of a linear probe into internal representations, these methods apply.

The new measures have all theoretical properties that a comparison measure should have as they are (pseudo-)metrics that do not depend on the overall scale of the representation (Williams et al., 2021). The new measures show some promise towards discriminating models as well, although this was not yet tested thoroughly. The new Bayesian measures fill the whole range of distances and show higher consistency across different image samples than the tested existing measures. A proper evaluation which methods work best under what circumstances is left for future work, because it should be based on many more models, neural data sets, comparison measures and stimulus sets.

Overall, the new methods are a great extension of our toolkit for comparing representations. The Bayesian methods are well justified on a theoretical level, provide novel statistical interpretations and applications, and can be computed effectively with little variance across image samples.

## Acknowledgments

This work was supported by the Luxembourg National Research Fund (FNR) under the project code BayesCompare.

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

# Appendix: Experiment details

**Test images** The experiments described in the paper were all performed based on the unlabeled images from MS COCO (Lin et al., 2015). They are provided by the COCO consortium at `http://images.cocodataset.org/zips/unlabeled2017.zip`. The annotation information for these images is licensed under a CC BY 4.0 license. The individual images were taken by a variety of flickr users and licenses under a range of different CC licenses. The license information for each image is available at `http://images.cocodataset.org/annotations/image_info_unlabeled2017.zip`.

**Networks** Three networks were used as implemented in the torchvision.models module version 0.15.2 (maintainers & contributors, 2016) with pytorch version 2.1.0 (Paszke et al., 2019).

The first network was Alexnet (Krizhevsky et al., 2012; Krizhevsky, 2014), for which the "IMAGENET1K_V1" weights were used, which were trained on ImageNet-1k (Russakovsky et al., 2014). For all convolutional and fully connected layers I used the representations after the non-linearity and included all layers in the comparison.

The second network was ResNet-18 (He et al., 2016) with "IMAGENET1K_V1" weights also trained on ImageNet-1k (Russakovsky et al., 2014). This networks architecture is primarily based on 4 main layers which each contain two residual blocks. I included all layers before and after the 4 layers and outputs of the two residual blocks for each main layer.

The third network was the Vision Transformer ViT-B-16 (Dosovitskiy et al., 2021) also with "IMAGENET1K_V1" weights. For this network, the 12 encoder layers outputs are analyzed. This network uses a slightly different preprocessing procedure than the other two networks, which causes the difference in preprocessed input locations in Figure 3.

**precompute** To speed up simulations, I precomputed the inner product matrix for all network layers for the first 1000 images from the image set and took subsets of those to compute the distances. All simulations below are based on this set of 1000 images. Precomputing these inner product matrices takes only a couple of minutes on the laptop, but removing the interaction with the neural networks from the analysis code simplifies it substantially.

## Example

For the example analysis presented in Figure 3, I computed the Jensen-Shannon divergence based metric for each pair of layers from all three networks and also compared to the pixel values of the preprocessed images. AlexNet and ResNet-18 use the same preprocessing, but the vision transformer uses a slightly different one, which causes two rows of divergences for the inputs.

To create the MDS embedding, I used sci-kit learn (Pedregosa et al., 2011) with the square root of the Jensen-Shannon Divergence as a precomputed distance metric.

## Choosing $a$

For choosing the signal to noise ratio or $a$, I looked at a range of distances and computed the distance with different numbers of images and signal to noise ratios. Two of those simulations are displayed in Figure 2. The images for this simulation were always the first $n$ images from the unlabeled images from MS COCO as described above. The displayed comparisons were chosen from the matrix of comparisons between all Alexnet layers and all ResNet-18 layers, but both displayed comparisons remained within ResNet-18.

## Comparisons between metrics

To compare to other metrics and analyze the stability of metrics 100 repetited analyses were run, each with a random image sample from the 1000 precomputed images. I varied the number of images using 25, 50 or 100 images. Figure 4 shows the average results for the 100 images case and Table 1 summarizes the standard deviations observed across the 100 repetitions. This analysis took the most computation of all experiments for this paper at 67 minutes of computation time on a MacBook pro M2Max laptop.

To enable direct comparisons, all measures of dissimilarity were implemented as functions of the inner product matrix $XX^T$ and transform them such that large values correspond to very different representations as follows:

**Jensen Shannon Divergence** was computed as described in the main text. For comparisons, a square root was applied to all values to yield a metric.

**Total Variation Distance** was computed as described in the main text and not transformed in any way.

**Centered Kernel Alignment** is naturally a function of the inner product / kernel matrices (Kornblith et al., 2019). Only linear centered kernel alignment is used here. The values were transformed by taking one minus the value such that large values correspond large differences instead of large alignment.

**Generalized Shape Metric** is a particular generalized shape metric that can be computed from the kernel matrix according to (Williams et al., 2021), described in their Appendix C.7. It is computed as the $\arccos$ of the centered kernel alignment. This conveniently already transforms the value into a metric with 0 corresponding to equivalence and 1 corresponding to the maximal distance. As noted by Diedrichsen et al. (2021), centered kernel alignment is equivalent to representational similarity analysis with a special whitened cosine measure for the similarity of dissimilarity matrices, giving another justification for this measure.

**Representational Similarity Analysis** is based on a dissimilarity matrix. Fortunately, euclidean distance is easy to compute from the inner product matrix: The squared euclidean distance distance from $x_i$ to $x_j$ is: $||x_i - x_j||^2 = (x_i - x_j)^T (x_i - x_j) = x_i^T x_i + x_j^T x_j - 2x_i^T x_j$. This formula to converts the kernel matrix into a squared euclidean distance matrix. Then, One minus the Pearson correlation of the upper triangular part of this matrix was used as the measure. This realizes one of many possible measures for similarity from representational similarity analysis (Walther et al., 2016).

**Representational Similarity Analysis:** $\arccos$ Here, we measure similarity using the cosine similarity of the upper triangular part of the distance matrix instead of the correlation (Diedrichsen et al., 2021). The resulting value was transformed into a distance measure by applying an $\arccos$ to the values. Effectively this process computes the angle between the distance vectors. Among other things, this makes this measure a metric on distance matrices, which induces a (pseudo-)metric on representations.

### Numerical stability

**Number of Images** For this analysis, I analyzed the variability across image samples from the comparisons between metrics.

**Number of Samples** To estimate the variance of estimates due to the sampling approximation, note that both approximations are simple averages over samples. Thus, the variance is simply the variance of the samples divided by the number of samples.

The first 100 images from the unlabeled set from MS COCO were used as described above, based on the precomputed inner product matrices and used $10\,000$ samples for each distance to compute the variance from the values. Computing these distances for all pairs of layers takes about 5 minutes on a MacBook pro M2Max.

Both metrics show a inverse u shaped relationship of variance on distance (Fig. 6). In the main text the maximum and mean of these variances are reported. It is clear from the scale that the estimation of total variation distances in substantially more accurate than the one of the Jensen-Shannon Divergence.

To compute a metric, we often compute the square root of the Jensen-Shannon Divergence, which looks problematic at first glance, because the square root function's derivative diverges at 0. One could fear that the variance of the distance (Fig. 6 C) diverges at 0. This means that we require an argument to show that this approximation converges and always has finite variance that converges to 0.

Formally, we can start with the statement that the distance estimate has finite variance. This is readily apparent because the distance estimate is always in $[0, 1]$ and thus must have variance $\leq 0.25$. Further, the mean of samples converges to the true value in probability. Thus, the distance also converges in probability to the correct value. Convergence in probability

means that we can choose an $\varepsilon$ range around the true value which will occur with high probability $1 - \alpha$ for all sample numbers $N \geq N_0$. For such $N$, the variance of the distance estimate must then be smaller than $(1 - \alpha)\varepsilon^2 + 0.25\alpha$, which we can make arbitrarily small by choosing $\alpha$ and $\varepsilon$ small enough. Thus the variance converges to 0.

Note, that these simple sampling approximations rely on us having analytic formulas for the densities. For other applications in machine learning these densities need to be approximated based on samples as well, which can severely bias these estimators (Murphy, 2022).

### Comparisons to NSD

The example analysis in Figure 5 is based on the natural scenes dataset (Allen et al., 2021) as prepared for the Algonauts challenge 2023 (Gifford et al., 2023), more specifically on voxels of the first subject in the left hemisphere reacting to the "places" functional localiser.

For the signal to noise ratio, a discrete prior with 10 different equally likely values placed logarithmically between $\exp(-5)$ and 1 was used. The overall evidence for a model is then the summed probability over these different parameter settings. The overall scale of the predicted covariances was set to $n$ times the voxel variance such that the predictions have the same variance as the voxels true responses.

### Analytic solution for the Posterior predictive

The Bayesian solution for a ridge regression is known (e.g. Wakefield, 2013; Schölkopf & Smola, 2002; Murphy, 2022). I give a short walk through here to demonstrate that the solution can be computed without handling covariance matrices in the original high dimensional representation space.

For $N$ observations of $k$ features and 1 output dimension, we have a matrix of representations in the model $X \in \mathbb{R}^{k \times N}$ and fit weights $\beta \in \mathbb{R}^k$ to predict a vector of outputs $y \in \mathbb{R}^N$ using the following model as in the main text:

$$\beta \sim N(0, \sigma_\beta^2 I) \tag{14}$$

$$y = X\beta + \varepsilon \qquad \varepsilon \sim N(0, \sigma_\varepsilon^2 I) \tag{15}$$

$$\Rightarrow y \sim N(X\beta, \sigma_\varepsilon^2 I) \tag{16}$$

For evaluation on neural data $X$ corresponds to the activations in a network layer, and $y$ are the measured neural activities in a particular measurement channel.

The Posterior for $\beta$ then is:

$$P(\beta|y) \quad \propto \quad P(\beta)P(y|\beta) \tag{17}$$

$$\propto \quad \exp\left(-\frac{1}{2}\left(\sigma_\beta^{-2}\beta^T\beta + \sigma_\varepsilon^{-2}(y - X\beta)^T(y - X\beta)\right)\right) \tag{18}$$

$$= \quad \exp\left(-\frac{1}{2}\left(\sigma_\beta^{-2}\beta^T\beta + \sigma_\varepsilon^{-2}\left(y^T y - 2y^T X\beta + \beta^T X^T X\beta\right)\right)\right) \tag{19}$$

$$\propto \quad \exp\left(-\frac{1}{2}\left(-2\sigma_\varepsilon^{-2}y^T X\beta + \beta^T(\sigma_\varepsilon^{-2}X^T X + \sigma_\beta^{-2}I)\beta\right)\right) \tag{20}$$

$$\propto \quad \exp\left(-\frac{1}{2}\left((\beta - \sigma_\varepsilon^{-2}\Lambda^{-1}X^T y)^T\Lambda(\beta - \sigma_\varepsilon^{-2}\Lambda^{-1}X^T y)\right)\right) \tag{21}$$

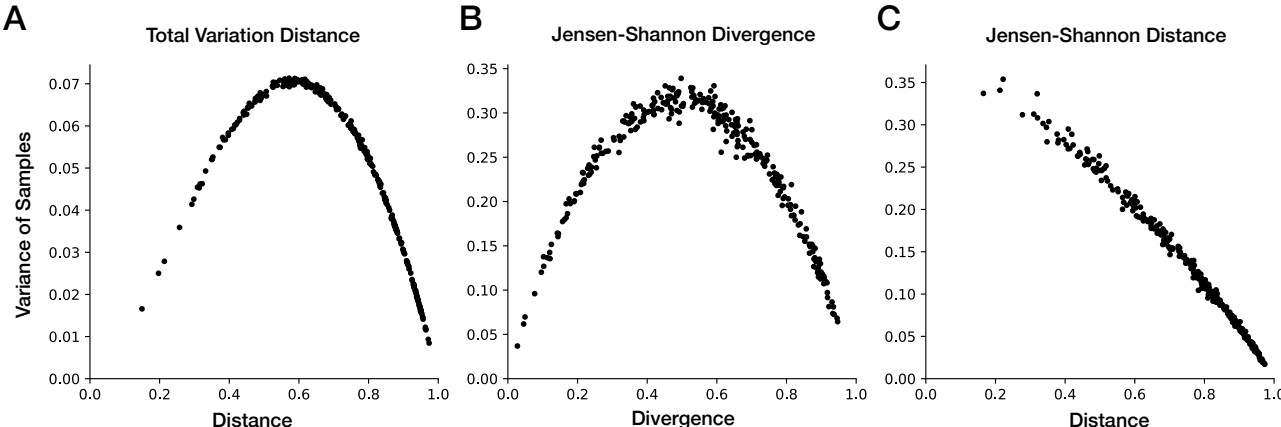

**A** Total Variation Distance

**B** Jensen-Shannon Divergence

**C** Jensen-Shannon Distance

Figure 6: Detailed numerical stability results on numerical stability of the sampling approximations. Variance of samples is the correct sum of variances such that the variance of the estimator becomes this value divided by the number of samples taken. The different points each correspond to one pair of layers and is estimated based on 100 images and 10000 samples. **A & B**: untransformed estimates of the total variation distance and the Jensen-Shannon Divergence. **C**: Jensen-Shannon *distance*, i.e. the square root of the Jensen Shannon divergence. For the distance this is the simple approximation computed by dividing the estimate by the squared derivative of the square-root transformation.

where $\Lambda = \sigma_\varepsilon^{-2} X^T X + \sigma_\beta^{-2} I$. Which is a normal distribution with mean and covariance matrix:

$$\mu_\beta = \sigma_\varepsilon^{-2} \left( \sigma_\varepsilon^{-2} X^T X + \sigma_\beta^{-2} I \right)^{-1} X^T y \quad (22)$$

$$\Sigma_\beta = \left( \sigma_\varepsilon^{-2} X^T X + \sigma_\beta^{-2} I \right)^{-1} \quad (23)$$

As we are dealing with a linear model, the posterior predictive for the mean of new data at $X'$ then is also a normal distribution with the following mean and variance:

$$X' \mu_\beta = \sigma_\varepsilon^{-2} X' \left( \sigma_\varepsilon^{-2} X^T X + \sigma_\beta^{-2} I \right)^{-1} X^T y \quad (24)$$

$$X' \Sigma_\beta X'^T = X' \left( \sigma_\varepsilon^{-2} X^T X + \sigma_\beta^{-2} I \right)^{-1} X'^T \quad (25)$$

Applying the Woodbury inversion formula to the Variance yields:

$$\Sigma_\beta = \left( \sigma_\varepsilon^{-2} X^T X + \sigma_\beta^{-2} I \right)^{-1} \quad (26)$$

$$= \sigma_\beta^2 I - \sigma_\beta^2 X^T (\sigma_\varepsilon^2 I + \sigma_\beta^2 X X^T)^{-1} X \sigma_\beta^2 \quad (27)$$

$$= \sigma_\beta^2 \left( I - X^T \left( \sigma_\varepsilon^2 \sigma_\beta^{-2} I + X X^T \right)^{-1} X \right) \quad (28)$$

Which yields the following formulas for the posterior predictive for the mean of new data:

$$\mu_{y'} = X' \sigma_\varepsilon^{-2} \sigma_\beta^2 \left( I - X^T \left( \sigma_\varepsilon^2 \sigma_\beta^{-2} I + X X^T \right)^{-1} X \right) X^T y \quad (29)$$

$$= \frac{\sigma_\beta^2}{\sigma_\varepsilon^2} \left( X' X^T - (X' X^T) \left( \sigma_\varepsilon^2 \sigma_\beta^{-2} I + X X^T \right)^{-1} (X X^T) \right) y \quad (30)$$

$$= \frac{\sigma_\beta^2}{\sigma_\varepsilon^2} \left( X' X^T - (X' X^T) \left( \sigma_\varepsilon^2 \sigma_\beta^{-2} I + X X^T \right)^{-1} (X X^T) \right) y \quad (31)$$

$$\Sigma_{y'} = X' \sigma_\beta^2 \left( I - X^T \left( \sigma_\varepsilon^2 \sigma_\beta^{-2} I + X X^T \right)^{-1} X \right) X'^T \quad (32)$$

$$= \sigma_\beta^2 \left( X' X'^T - (X' X^T) \left( \sigma_\varepsilon^2 \sigma_\beta^{-2} I + X X^T \right)^{-1} X X'^T \right) \quad (33)$$

These formulas contain only matrices with size $N \times N$, none of size $k \times k$, such that the only computation that depends on the number of features in the representation is computing the original inner product matrix.

