# OpenReview forum: "Bayesian Comparisons Between Representations"
_ccneuro.org/CCN/2025/Proceedings — CCN 2025 Proceedings asProceedingsPoster_

### Official Review · Reviewer_Xcm1 · 2025-03-29
**Lack of motivation and novelty**

**Soundness:** 1
**Clarity:** 2

**Comments:**

### Interest

While the problem of representational similarity is of great interest to CCN community, I am concerned about the novelty of the work.

### Soundness

Many critical and foundational citations which are closely related to the proposed metric are omitted. There is no clear motivation behind the introduction of these “new” metrics and there are no clear advantages of these metrics compared to the existing methods.

### Clarity

The paper is clearly written and easy to follow.

## Comments

### Summary

Based on the claim that Bayesian statistics is well suited for representational similarity *“because they deal better than frequentist statistics with small datasets and parameters that are not strongly constrained by the data”,* the author proposes comparing distances between induced prior distributions over the functions constructed by linear readouts.

### Concerns/Lack of Citations

- The proposed linear readout model as a Bayesian prior is not a new idea has been developed for deep neural networks generically [https://citeseerx.ist.psu.edu/document?repid=rep1&type=pdf&doi=db869fa192a3222ae4f2d766674a378e47013b1b] and deserves citation in my opinion.

- Applying distances between probability distributions to comparing representations is not a new idea and has been used extensively [e.g. for training generative models https://arxiv.org/abs/1606.04218]. Such distances were rigorously studied for general kernel embedding spaces [https://arxiv.org/pdf/0901.2698, https://www.jmlr.org/papers/volume13/gretton12a/gretton12a.pdf] and generalize beyond the two metrics studied in the paper.

- The full description of inductive biases and generalization of these priors were mentioned in the introduction section but were never revisited. Inductive biases of regression with linear readout models and their generalization properties are already well understood without the Bayesian framework [https://arxiv.org/abs/2002.02561, https://arxiv.org/pdf/2102.07238], and it is not clear how this new Bayesian approach encodes these inductive biases differently.

- The approximations in Eq.(7) and Eq.(11) are heavily biased estimators that require careful analysis to make them unbiased.

The experiments performed on various neural networks, such as ViT-B-16, have been carried out by many other researchers and should be compared with results from different studies.

### Questions

- In Figure.4, TVD yields the same distance for different CKA distances. What does this imply? Does CKA underestimate distances or is TVD insensitive to some representational aspects that CKA can capture?

- While CKA, GSM, RSA, and RSA-arccos qualitatively look similar, they are very different from the proposed measures. What is the reason?

**Expertise:**

3

**Interest:**

2

---

> ### Author Rebuttal · Authors · 2025-04-15
>
> I did not intend to give the impression that I am the first to apply distances on probability distributions in machine learning and do agree that the applications for generative models are a good cross-reference to give to the reader. Accordingly, I have added a paragraph to the related works section that makes this connection.
>
> In none of the work I know and none of the papers referenced by the reviewer here, such distances are used to make comparisons between internal representations of neural networks though. The application of comparing the predictions of generative models to each other and/or to sampled real data remains quite distinct from the comparisons I propose in this manuscript.
>
> ## Biased approximations
>
> In this case these sampling approximations are unproblematic. This judgement might be based on the wrong impression that at least one density is based on the samples as well. Here, we can compute both $p_1$ and $p_2$ analytically which makes the approximation of these integrals a much simpler problem for which the sample means are an unbiased estimator. I confirmed this for cases where we can compute analytic solutions.
>
> ## Comparisons
> I do include results for different metrics in Figure 4. I do not discuss the results in detail, because I want to avoid too much focus on the fairly arbitrary examples I present. The point is that we can draw such comparisons for neural networks at full scale and get interpretable, seemingly sensible results. To draw scientific conclusions we will need to apply the same procedures to many more DNNs.
>
> **Q1**
> I do not think this observation implies any of the statements the reviewer proposes. A particular metric judging two distances to be the same does not in any way imply that the result of a different metric should be the same, too. This merely reflects that CKA and TVD weigh differences in the space of representations differently, i.e. that they are not identical. I discuss the connections for distances of 0 in section “Equivalent representations”.
>
> **Q2**
> The correlations between the other measures among themselves are not actually higher than to the new measures I propose here, i.e. similar plots as Fig. 4a show some scatter for any pair of the other measures, too. My measures yield higher distances on average. This is not an important difference though as the overall scale of a distance is quite meaningless can be reduced for the new measures by reducing the signal to noise ratio.

---

### Official Review · Reviewer_FH66 · 2025-04-01
**A very well-thought out paper**

**Soundness:** 3
**Clarity:** 2

**Comments:**

The framework is well thought-out and also explained in detail.

Interest: The work is of clear interest to the CCN community because there has been a lot of recent focus on representation similarity metrics. This method also provides an uncertainty estimate, which makes it a desirable method.

Soundness: I believe most of the math checks out and the conclusions in the manuscript seem justified to me.

Clarity: The author explains everything in detail, walking through their rationale. However, I did feel that people not well-versed in Bayes might struggle a little bit in understanding this paper.

Other comments: I believe the results would benefit from comparing with what other metrics tell us, for example, on line 449, "The best performing layers at predicting this area are Conv5 from AlexNet and Block 4_1 from ResNet-18".  Is this consistent with what other metrics predict?

Fig 1: The dotted line in subplot B should be described in the figure caption (I am assuming it corresponds to the value of 1, but there are no numbers on the x-axis to confirm that.) Please add something in the Figure that illuminates how you got the distributions in 1B.

Typo on line 95: ‘mayor’ approaches instead of ‘major’

Line 114: “Comparisons based on fitting an explicit map are asymmetric by default.“
While this is true, there are instances in the literature of symmetrizing regression-based comparisons by fitting one model from representation 1 to representation 2, another from representation 2 to representation 1 and then taking the mean of the scores obtained through the two fits. (I believe Soni et al., 2024 used a symmetrized linear score).

Typo on line 141: “metrics depends” should be 'metric depends'

Typo on Line 388: ‘intends and purposes' should be intents and purposes

What do we learn from Fig 3B? Please call out and describe the individual figs in the text, instead of just the global figure

**Expertise:**

2

**Interest:**

3

---

> ### Author Rebuttal · Authors · 2025-04-15
>
> Thank you for your positive evaluation! I addressed most of your comments in the revision.
>
> **Line 114 Soni et al.**
> If I found the right Soni (Conclusions about Neural Network to Brain Alignment are Profoundly Impacted by the Similarity Measure) they actually run both directions separately as two scores, but I softened the wording nonetheless.
>
> ## General motivation
> As I have some space left in my response to this reviewer, I want to answer the concern the other two reviewers had about more  advantages of the metrics I propose, to confirm that my new proposal is indeed helpful in practice.
>
> The main advantages of the comparisons I propose here are their theoretical connections, which enable substantial simplifications for further analysis:
> - A proper Bayesian analysis of the read out predictions opens the door towards a coherent statistical analysis of the models without requiring any dimensionality reduction, cross-validation, bootstrapping, etc. Removing these complications could make analysis faster and simpler, more powerful due to using the whole dataset for evaluation, and would leave fewer analysis choices to the researcher improving coherence of the field.
> - The TVD and JSD measure discriminability. Thus, they are an excellent direct measure of power for experimental design. For example, we can judge whether a particular experiment was powerful enough to differentiate a pair of models. This is valuable information to judge experiments.
> - The measures I propose are ideal targets to optimise controversial stimuli as the JSD and TVD measure fairly directly how discriminable models are and we can compute them conditioned on existing data such that we can update our stimulus selection based on existing data.
> - And for machine learning applications a metric between representations that can be computed quickly including a stochastic gradient, that is bounded to the unit interval and that we can tune to be more or less sensitive seams like a helpful addition, too.
>
> I have revised the discussion substantially to make these advantages clearer in the manuscript.
>
> I also agree with the reviewers that additional direct comparisons of the methods to existing ones would be valuable. We are working on thorough empirical evaluations of these methods and on applying the theoretical connections I describe, but these evaluations are not complete yet and will alone fill more space than we could possibly have in this paper.

---

> > ### Comment · Reviewer_FH66 · 2025-04-18
> >
> > Thanks for the response! I believe my concerns have been answered, except:  "Please add something in the Figure that illuminates how you got the distributions in 1B."
> >
> > I still recommend accepting the paper.

---

### Official Review · Reviewer_yQQF · 2025-04-01
**A Bayesian treatment of representational similarity**

**Soundness:** 2
**Clarity:** 1

**Comments:**

Thank you for this thought-provoking piece of work!  This paper formulates a metric on representations in terms of Bayesian statistics, in the sense that representations are judged on the ability to distinguish the distributions of random linear readouts from the representations from samples.  This is a valuable contribution to the area of representational similarity metrics, which are often motivated by geometric or empirical intuition.  However, I have some questions and concerns related to this manuscript.

The first concern is about the number of free parameters and choices are required to formulate this metric.  In my understanding the choices made to formulate the Bayesian readout metric are:
- choice of readout function
- choice of weight prior distribution
- choice of signal and noise variance, or free parameter $a$
- choice of metric on probability distributions

There is some justification for each of these choices, which is great, but can the author comment on how sensitive scientific conclusions might be to these choices?  For example, if I use a different prior, or a different heuristic for choosing $a$, how much do your results comparing deep network layers change?  Are these results very sensitive to each of these choices?  I am a bit concerned by these the dependence of these metrics on the number of sample images and how this is fixed using a heuristic to set $a$.  To what extent can I tune $a$ and $b$ to change the scientific result?  How should I set $b$?

Second, I would like the authors to comment on the scientific value of a metric if they "broadly agree with existing metrics, but are more stringent"?  Do these metrics help solve an existing problem with previously existing distance metrics, or is the main contribution here that they are more rigorously motivated?  I appreciate the theoretical construction here is valuable, but I am wondering about any practical insight given by this work that would not be observed using other metrics.  Figure 4 gives a nice comparison between the Bayesian metric and some other metrics that are generally used, but I am not sure how to interpret this figure.  I can see that these metrics do not have one-to-one relationships, but how does a practitioner use this or any other analysis to judge which metric is meaningful?  You say that in broad strokes the metrics all agree in the sense that close-by neural network layers tend to be similar to each other, some of the intermediate conv layers produce similar representations, and the final readout layers are very different from the conv layers.  Does this mean that it does not matter which metric a practitioner chooses?  I think the paper could benefit from commentary on this.  The last sentence of the paper claims that these metrics "work well in practice", but I am not sure how the author is judging this or how we should conclude that these metrics work better than existing methods.


Minor concerns:

- Equation 6, A is not formally defined
- How did you handle the flattening of the convolutional layers?
- Figure 3:  what does this look like for randomly initialized networks?  (in particular for the ViT)
- Table 1:  I don't see where n is defined. What is this column indicating?
- Line 463:  what is the place localizer and why are you using the first voxel that corresponds to it?
- Discussion:  I don't see how this metric nicely unifies existing methods for comparing representations, except in the sense that it involves linear kernel matrices, and so do other methods.  As far as I can tell you haven't argued that there is any mathematical relationship between these metrics and the other metrics that take kernel matrices as the arguments.  In fact figure 4 shows that they do not have a trivial or 1-to-1 relationship with CKA, Generalized Shape Metrics, or RSA methods.
- Line 501:  Could you clarify what you mean by "the analyses presented here apply a separate analysis for each voxel"?  Does this mean you are fitting one linear readout per voxel and per layer?  Can you relate the neural data situation to the mathematical setup at the beginning to make this easier on the reader (i.e. tell me what corresponds with X, beta, and y and how you chose the signal to noise ratio).
- https://arxiv.org/abs/2411.08197 also considers the connection between representational similarity measures and linear readouts, and should probably be cited in the related work.


Typos/grammar:

line 110:  a recent method

line 180:  are degenerate

line 235: grammar

Figure 2 caption:  line 6, typo "divergence between two"

line 388:  for all intents and purposes

line 468: knowledge of how

**Expertise:**

3

**Interest:**

3

---

> ### Author Rebuttal · Authors · 2025-04-15
>
> Thank you for your kind words!
>
> ## First concern
> This is a valid concern that my methods share with the existing methods that also allow many choices. Other read out functions and other priors on the weights are indeed likely to produce different conclusions, but the simple linear read out with a Gaussian prior is clearly the simplest solution and should be our default choice. I think this will result in similar outcomes as for centered kernel alignment which does give different results for different kernels, but in practice the linear kernel is used almost exclusively.
>
> For the signal and noise variances, I have two separate answers:
> - For evaluations on neural data the signal and noise variances are free parameters that we should fit to the data.
> - For comparisons between model representations, we could get different results by changing the relative scale. To check, I ran my measures with different signal to noise ratios through the same procedure I used for comparing measures in Figure 4. For a factor 5 lower signal to noise ratio, the spearman rank correlation was 0.982 for the TVD and 0.980 for the JSD. This is not perfect, but a very high correlation that will rarely allow different scientific conclusions.
>
> Finally, the choice of metric on probability distributions: The TVD and JSD give very similar results as displayed already in the original manuscript. So for the ones I am proposing here, this choice appears to be largely irrelevant.
>
> So, in practice, I think we will focus on the linear model with Gaussian priors and the other choices fortunately do not change the fundamental scientific conclusions we would draw much.
>
> ## Second concern
> This comment prompted me to revise the discussion substantially. I also respond here a bit more in my rebuttal to reviewer FH66 where I had some space to do so. Hopefully the reviewer will find these responses address their concern.
>
> ## Minor concerns:
>
> **Discussion**
> As I generally overhauled the discussion, I replaced this statement with the more accurate point  that there are interesting theoretical connections to both the linear encoding models and the kernel or distance based methods.
>
> **analysis per voxel**
> I do evaluate a separate readout per voxel and per layer. Given the direct formula for the predictive distribution, there is no real fitting here. For the SNR, I used 10 discrete values with a uniform prior & fixed the predicted variance to be the same as the true variance. See appendix for details.

---

> > ### Comment · Reviewer_yQQF · 2025-04-19
> >
> > Thank you for your explanations and revisions.  I recommend accepting this paper.

---

### Meta-Review · Area_Chair_akpF · 2025-05-03

**Ccn Recommendation:** Accept as Proceedings

**Metareview:**

Reviewer 1 was overall positive but raised that (1) the methods have many degrees of freedom concerning analytical choices, such as which readout functions are used, and (2) many other methods exist which yield results that are in part highly correlated with this approach. The reviewer also seemed to feel that the method still represents an advance and that the author engaged with their review in a way that led to a meaningful adaptation of the manuscript.
Reviewer 2 was very positive (interest broad, soundness strong, clarity adequate). The clarity point reflects that s/he wrote, "I did feel that people not well-versed in Bayes might struggle a little bit in understanding this paper." Most other points were minor and largely addressed by the author.
Reviewer 3 was more skeptical, raising concerns about lack of motivation and novelty, and in particular that "(m)any critical and foundational citations which are closely related to the proposed metric are omitted." A more substantial critique was that some approximations are biased estimators. The author responded by incorporating the relevant literature. Concerning the biasedness, the author makes a convincing case that it is not problematic. All minor issues are addressed as well.

Overall, the paper receives mostly positive ratings and engaged in the review process so as to address major concerns. I therefore recommend acceptance to the proceedings.

**Summary:**

The paper presents a new method for comparing the predictions of neural networks for a given dataset. The author proposes to use  predictive distributions of linear readouts of hidden representations and compare multiple models using the Jensen-Shannon distance between these to compare models.
The paper was reviewer by 3 reviewers. All reviewers had sufficient expertise and were consistently very positive about how much interest this paper would attract, positive about the soundness and mixed about clarity. The main issue that was raised by all reviewers concerned the motivation of the paper given numerous existing approaches. R1 and R2 found the review process constructive and recommend accepting the paper. R3 did not reply to the response by the author, but from my point the author addressed these criticisms sufficiently, mostly by adapting the discussion.

**Expertise:**

2